# Bases for the Adequate Development of Nutritional Recommendations for Patients with Inflammatory Bowel Disease

**DOI:** 10.3390/nu11051062

**Published:** 2019-05-12

**Authors:** Esteban Sáez-González, Beatriz Mateos, Pedro López-Muñoz, Marisa Iborra, Inés Moret, Pilar Nos, Belén Beltrán

**Affiliations:** 1Inflammatory Bowel Disease Unit, Gastroenterology Department, La Fe University and Polytechnic Hospital, 46026 Valencia, Spain; esteban.digestivo@gmail.com (E.S.-G.); bmateosa@gmail.com (B.M.); pedro.lopez8928@gmail.com (P.L.-M.); marisaiborra@hotmail.com (M.I.); Ines.Moret@uv.es (I.M.); pilarnos@gmail.com (P.N.); 2Inflammatory Bowel Disease Research Group, Medical Research Institute Hospital La Fe (IIS La Fe), 46026 Valencia, Spain; 3Biomedical Research Network Center for Liver and Digestive Diseases (CIBEREHD), 28029 Madrid, Spain

**Keywords:** diet, inflammatory bowel disease, microbiota, intestinal barrier, nutrients, immunity

## Abstract

Inflammatory bowel disease (IBD) is a chronic and relapsing inflammatory condition of the gastrointestinal tract; it is a heterogeneous and multifactorial disorder resulting from a complex interplay between genetic variation, intestinal microbiota, the host immune system and environmental factors such as diet, drugs, breastfeeding and smoking. The interactions between dietary nutrients and intestinal immunity are complex. There is a compelling argument for environmental factors such as diet playing a role in the cause and course of IBD, given that three important factors in the pathogenesis of IBD can be modulated and controlled by diet: intestinal microbiota, the immune system and epithelial barrier function. The aim of this review is to summarize the epidemiological findings regarding diet and to focus on the effects that nutrients exert on the intestinal mucosa–microbiota–permeability interaction. The nature of these interactions in IBD is influenced by alterations in the nutritional metabolism of the gut microbiota and host cells that can influence the outcome of nutritional intervention. A better understanding of diet–host–microbiota interactions is essential for unravelling the complex molecular basis of epigenetic, genetic and environmental interactions underlying IBD pathogenesis as well as for offering new therapeutic approaches for the treatment of IBD.

## 1. Introduction

Inflammatory bowel disease (IBD) is a chronic and relapsing inflammatory condition of the gastrointestinal tract. Crohn’s disease (CD) and ulcerative colitis (UC) are the two principal types of IBD. Its prevalence has been increasing worldwide, with the highest incidence found in Western countries [1]. The precise etiology of IBD remains unclear; however, interactions between genetic, microbiotic and environmental factors are associated with its pathogenesis [2]. There is a compelling argument for environmental factors such as diet playing a role in the cause and course of IBD [3], given that three important factors in the pathogenesis of IBD can be modulated and controlled by diet: intestinal microbiota, the immune system and epithelial barrier function.

Patients with IBD tend to present with malnutrition, especially when they are having a flare or have chronic, only partially controlled, intestinal inflammation. Currently, to deal appropriately with malnutrition is considered a clinical quality criterion, especially for preventing the complications that can arise under a malnutrition situation (e.g., infections, postsurgical complications, immunity alterations). Thus, there is a tendency to at least consider the patient’s nutritional status before surgery [4], and a guide has recently been published with dietary recommendations [5]. However, the gastroenterologist typically gives little advice to patients with IBD regarding daily diet or nutrient considerations. Patients are often told, “eat what you can tolerate,” and discussion on diet is typically very limited. Despite the scarce attention gastroenterologists give to nutrients, approximately 40% of patients with CD believe that diet can control symptoms, and approximately 80% believe diet is important in the overall management of the disease [6]. Some 40% of patients with IBD have attempted various diet therapies, often without the assistance of a physician or dietician [7].

Perturbations related to dietary intake are thought to relate to the consumption of a dietary pattern that negatively alters gut microbiota composition and intestinal permeability [8]. Recent experimental evidence has suggested a crucial factor of barrier dysfunction in the onset of IBD [9]. The aim of this review is to summarize the epidemiological findings regarding diet and to focus on the effects that nutrients exert on intestinal mucosa–microbiota–permeability interaction.

## 2. Diet Influences Inflammatory Bowel Disease: Epidemiological Clues

Epidemiological studies have shown associations between the intake of specific dietary components and the risk of developing IBD [10,11,12,13,14,15,16,17]; however, association does not necessarily imply causality. These studies must be considered with caution, given that methodological inconsistencies and a recall bias could be affecting the results observed. Furthermore, in a complex disease with dietary triggers, it can be difficult to unravel the role of individual dietary risk factors because dietary patterns often involve exposure to clusters of nutrients. These clusters are favored by the industrial processing of food.

Classically, no environmental factor has been claimed to impact directly on the pathogenesis of IBD, apart from tobacco [18]. However, a recent systematic review and meta-analysis have shown that breastfeeding provides a protective factor against the development of CD and UC, both in pediatric and adult-onset disease. This inverse association is consistent in studies worldwide and supports the recommendation of breastfeeding in infancy to reduce the risk of IBD development [19]. Further research on the effects of breast milk on the microbiome and modulation of the innate immune system are needed.

A systematic review of the literature, initially including 2085 publications, was published in 2011 [20]. Nineteen studies fulfilled the inclusion criteria, containing a total of 2609 patients with IBD (1340 UC and 1269 CD). The authors concluded that a high intake of total fats, polyunsaturated fatty acids (PUFAs), omega-6 fatty acids and meat were consistently associated with increased risk of developing UC as well as CD. High vegetable intake was consistently associated with decreased risk of UC, whereas fiber and fruit intake were consistently associated with reduced risk of CD.

Dietary fats are associated with an increased or decreased risk for IBD, depending on the type of fat. Among 170,805 women followed over 26 years, a high intake of dietary long-chain omega-3 PUFAs was associated with a reduced risk of UC. In contrast, high intake of trans-unsaturated fats was associated with an increased risk of UC [15]. Various data have been extracted from an analysis of 366,351 individuals with IBD from the European Prospective Investigation into Cancer and Nutrition study cohort. Globally, a dietary imbalance with a high consumption of sugar and soft drinks and a low consumption of vegetables has been associated with UC risk [12]. Regarding fats, a sub-study analyzing 126 incident cases of UC showed that the patients ranking in the superior quartile of linoleic acid intake had a higher risk of developing UC [21]. Similarly, another analysis considering 70 incident cases of CD has shown that higher intake of the omega-3 PUFA docosahexaenoic acid prevents CD development [22]. A recent population-based study of environmental risk factors shows an increased risk of CD with frequent fast-food consumption (Western diet) before the diagnosis [23]. A Western diet typically contains high levels of omega-6 PUFAs and low levels of omega-3 PUFAs. The ratio between the intake of omega-6 and omega-3 PUFAs is considered a risk factor for IBD.

Data from the Nurses’ Health Study including 170,776 participants showed that higher long-term fruit intake was associated with a lower risk of CD but not UC [15]. Multiple studies have reported a negative association between dietary fiber intake from fruits or fruits and vegetables and subsequent risk of CD. Fiber is the dietary component with the greatest agreement in epidemiological studies. In a recent prospective study [24], fiber from fruit has also been implicated in preventing the establishment of pouchitis in patients with UC. It is important to remember that the fiber implicated in these effects is soluble fiber, which is fermentable by bacteria. Soluble fiber is usually less present in food; however, the proportion of insoluble to soluble fiber could be the factor limiting tolerance to various foods.

Animal proteins have also been claimed to increase the risk of IBD [10]. A prospective study (E3N) that included 67,581 French women showed that high animal protein intake was associated with a significantly increased the risk of IBD, particularly with UC. However, other smaller studies have not confirmed this risk [20]. In a separate study evaluating dietary intake and relapse of UC, red meat was found to have the strongest association (odds ratio 5.19; 95% confidence interval 2.09–12.9) with relapse [14].

Food additives have been progressively increasing in processed food. Both sweeteners and emulsifiers, used to enhance the texture and stability of foods, have been shown to disrupt host–microbiota interaction and to facilitate intestinal inflammation [25]. However, it is difficult to ascertain whether the additives are risk factors because most of the questionnaires used to assess dietary intake do not evaluate food additives.

Currently, it seems clear that dietary patterns are more important than individual foods for considering the risk of IBD. The consumption of vegetables, fruits, olive oil, fish, grains and nuts (prudent diet) was associated with a decreased risk of developing CD in both boys and girls in a Canadian pediatric study. A Mediterranean dietary pattern is high in extra-virgin olive oil, vegetables, fruit, legumes, nuts and seeds, with a moderate consumption of fish, poultry and milk products, and is low in processed foods, baked goods and red and processed meat. This dietary pattern is high in monounsaturated fats, omega-3 PUFAs, fermentable fiber and polyphenols, and adherence to a Mediterranean dietary pattern is associated with lower levels of inflammation biomarkers [26].

## 3. Impact of Nutrients on Intestinal Permeability, Microbiome and Immunity

Dietary fiber is not digested or absorbed by host cells, given that mammalian cells largely lack the necessary enzymes to degrade them. Instead, dietary fiber is subjected to bacterial fermentation in the gastrointestinal tract. Although a wide range of bacteria ferment dietary fiber, each bacterium has a substrate preference based on its enzymatic activity. Thus, dietary intervention can remodel the gut microbial composition by customizing the content of dietary fiber [27]. Short-chain fatty acids (SCFAs), such as acetate, propionate and butyrate, are the major end products of microbial fermentation of dietary fiber. They are key energy substrates used by colonocytes [28]. Butyrate enhances intestinal epithelial barrier function via hypoxia-inducible factor-1, which regulates the integrity of epithelial tight junctions [28,29,30]. It also increases the synthesis of the MUC2 protein, the main component of intestinal mucus [31,32]. Thus, a lack of dietary fiber might compromise epithelial integrity and mucus production due to insufficient SCFA generation, resulting in impaired intestinal barrier function [33]. Recently, it has been observed that consumption of a low-fiber diet leads to the disruption of intestinal barrier function through a mechanism that is independent of SCFAs. In the absence of dietary fiber, some commensal bacteria use host mucus glycans to meet their energy needs [34]. As a result, these mucolytic bacteria become the predominant species within the gut microbiota. Importantly, a bloom of mucolytic bacteria results in the degradation of the colonic mucus layer that renders the host susceptible to enteric pathogens [34] (Figure 1). These changes facilitate bacterial adherence and translocation into the epithelium, which introduce immunomodulatory effects. In summary, a diet poor in soluble fiber can condition both dysbiosis and a subsequent decrease in the mucus layer, which increases the permeability already affected by the fiber-poor diet.

The gut microbiota is thought to play a crucial role in human health and prevention of disease through a variety of mechanisms, including the production of short-chain fatty acids (SCFAs), which are important for maintaining gut homeostasis and optimal immune function. Ingested fiber can influence fecal microbiota profiles, cause changes in the complex gastrointestinal environment and promote the growth of bacteria in general and potentially beneficial bacteria in particular.

Fat- and refined sugar-rich diets can condition a low production of butyrate, which has been described in patients with CD, together with a diminution in the butyrate receptor [35] Butyrate plays an important role in downregulating inflammation by suppressing transcription of cytokines and increasing differentiation and the population of lamina propria Tregs [33,35,36]. Butyrate can also enhance innate immunity by upregulating defensins and cathelicidins in animal models [36]. Defensins have been reported to be permanently decreased in patients with IBD, and whether a dietary intervention could reverse this condition deserves further investigation. High-fat diets (without high sugars) have been shown to increase tumor necrosis factor (TNF)-α and interferon-γ expression, and to decrease levels of colonic Tregs [37].

Dietary amino acids play important roles in gut homeostasis. Amino acid metabolic profiles in the blood, urine, feces and intestinal tissues are also altered in patients with IBD and correlate with the severity of the disease. Additionally, metagenomic studies have revealed that amino acid biosynthesis genes are downregulated and amino acid transporter genes are upregulated in the gut microbiome of patients with IBD, indicating that the gut microbiota lessens the production of amino acids and increases the rate of their use [38,39]. Certain amino acids are critical for immune T-cell function as well as for the proliferation of macrophages [40,41]. Thus, the demand for certain amino acids by host cells and the gut microbiota can increase as a consequence of inflammation.

Tryptophan is an essential amino acid that can be found in fish, meat and cheese. Its metabolites, such as kynurenine, indole-3-aldehyde and indole-3-acetic acid, can act as ligands for the aryl hydrocarbon receptor, a critical regulator of immunity and inflammation involved in adaptive immunity and intestinal barrier function [42,43,44]. Indoleamine 2,3 dioxygenase-1 (IDO1) is ubiquitously expressed in epithelial cells, dendritic cells and macrophages. IDO1 is the first step in the kynurenine pathway, a major route for tryptophan catabolism. IDO1 regulates the differentiation and maturation of adaptive immune cells [45]. Kynurenine is an initial metabolite of IDO1-mediated tryptophan catabolism and the kynurenine/tryptophan ratio is a surrogate marker of IDO1 activity. A recent clinical study has shown that serum tryptophan levels are lower and the kynurenine/tryptophan ratio is elevated in patients with IBD compared with healthy controls [15]. Additionally, indoleamine 2,3-dioxygenase 1 (IDO1) mRNA expression in colonic tissues is significantly higher in IBD and correlates with disease severity, suggesting the kynurenine pathway is upregulated in IBD.

Arginine and glutamine are two semi-essential amino acids that can impact gut homeostasis and innate immunity. Both are diminished in prospective cohorts of patients with IBD, and both are implicated in wound repair mechanisms [46]. Furthermore, arginine regulates macrophage differentiation into the M1 or M2 phenotype following environmental cytokine guidance, whereas glutamine reduces oxidative stress and cytokine production via downregulation of the NF-kB and signal transducer and activator of transcription proteins (STAT) signaling pathways.

The upregulation of amino acid metabolic pathways is one of the major inflammation-related metabolic shifts in the host and the microbiota. In other words, host cells and resident microbes have an increased demand for certain amino acids in the context of IBD. Thus, dietary supplementation of these amino acids is a key strategy for the treatment of IBD. However, it is noteworthy to mention that other amino acids can fuel pro-inflammatory responses in IBD, and restriction of certain amino acids can attenuate intestinal inflammation. A more thorough understanding of the specific roles various amino acids play in IBD and supplying the optimal amount of anti-inflammatory amino acids while limiting the consumption of proinflammatory amino acids can lead to the development of effective amino acid-based dietary interventions.

## 4. Mesenteric Fat and Epigenetic Considerations

Today, patients with IBD sometimes have malnutrition with obesity, associated with metabolic syndrome. Obesity has also been associated with the development of more active IBD, with a higher demand for therapeutic resources. Patients with IBD, especially CD, present fat-wrapping or “creeping fat,” which corresponds to ectopic adipose tissue extending from the mesenteric attachment and covering the majority of the small and large intestinal surface. Mesenteric adipose tissue in patients with IBD presents several morphological and functional alterations; e.g., it is more infiltrated with immune cells such as macrophages and T-cells. This fat tissue is not an “innocent bystander,” but actively contributes to intestinal and systemic inflammatory responses [47]. It overexpresses peroxisome proliferator-activated receptor gamma (PPAR-γ), which is a “master” regulator of the adipogenesis program and the major orchestrator of visceral adipose tissue Treg accumulation, phenotypes and function. Mesenteric adipose tissue is colonized by luminal bacteria, and PPAR-γ expression is modulated by several bacterial stimuli. Thus, the presence of metabolically active adipose tissue should be taken into consideration when attempting to change the alterations facilitating bacterial adherence and/or clearance and permeability regulation of innate/adaptive immunity.

Similarly, epigenetic modulation of gene expression should be contemplated when considering how other environmental factors can affect the mucosa–microbiome–immunity interplay. A Westernized high-fat diet, full of refined carbohydrates, is strongly associated with the development of IBD, contrary to a diet high in fruit, vegetables and omega-3 PUFAs, which is protective against it [26]. Epigenetic mechanisms could explain the connection between genes and environmental factors in triggering the development of IBD. The definition of epigenetics refers to heritable alterations of gene expression events that are caused independently of genetic information carried by the primary DNA sequence. The main epigenetic mechanisms controlling gene expression include DNA methylation, histone modifications and small and long noncoding RNAs [48]. The main objective of epigenetics is to modulate gene expression without modifying the basic DNA sequence. All the mechanisms involved in epigenetics are involved in correct cell development, differentiation, function and homeostasis. Moreover, these mechanisms are influenced by exposure to environmental factors, persist through mitosis and meiosis and, more importantly, can be reversed. Recent advances have indicated that epigenetic variation has an important influence on interactions between nutrients and the genome, which modifies disease risk [49,50]. Diet is known to influence epigenetic changes associated with disease and to modify gene expression patterns in a state of disturbed immunity. A number of nutrients have been shown to modulate immune responses and can potentially counteract inflammatory processes [51]. Some studies have suggested that secondary plant metabolites, such as polyphenols, might modulate gene expression, chromatin remodeling and DNA methylation [52]. Epigenetic effects have also been shown for other dietary components, such as curcumin [53]. Moreover, specific components of the Mediterranean diet, particularly nuts and extra-virgin olive oil, have recently been found to be able to induce methylation changes in several peripheral white blood cells, showing a role for specific fatty acids in epigenetic modulation [54]. Thus, considering the status of epigenetic mechanisms in particular patients, it is important to understand whether a dietary intervention could help or not.

## 5. Diet as Possible Therapy

Despite the fact that a growing body of knowledge related to diet is emerging in terms of IBD pathogenesis, there is no strong evidence for dietary interventions helping to induce remission or to maintain response. Dietary therapies, such as enteral exclusive nutrition (EEN), have traditionally and primarily been used to induce remission in early or new-onset CD. EEN has been shown to be effective for inducing clinical remission, improving nutritional status, improving body composition, inducing mucosal healing and decreasing proinflammatory cytokines in pediatric and adult patients with CD. EEN has been extensively used for induction of remission in pediatric CD, in which avoidance of steroids is critical for childhood growth. However, EEN has been less often used for adult patients. Although some studies have assessed EEN in adults, most were less conclusive and had multiple confounding factors; thus, this practice is uncommon in adults with CD [55]. EEN involves the exclusive use of liquid nutrition in the form of medical formulas, without exposure to other foods, usually for 6–8 weeks. This effect does not depend on the protein source (a type of formula), but it does depend on the exclusion of ordinary table food [56]. Several recent pediatric studies have demonstrated and confirmed that EEN can induce remission in 60%–86% of children, accompanied by a significant decrease in inflammatory markers, such as the erythrocyte sedimentation rate, C-reactive protein and fecal calprotectin [57,58,59]. This recent body of literature has evaluated the long-term benefits associated with the use of EEN in mild-to-moderate pediatric CD. In comparison with the use of steroids, and corrected for disease severity, EEN has been associated with higher remission rates, better growth and longer steroid-free periods, but it did not differ in regard to other outcomes such as relapse. The benefit of EEN was lost when partial enteral nutrition (PEN) was used instead, with access to a free diet granted to patients [60]; thus, reaffirming the principle of exclusivity, suggesting that the mechanism depends on the exclusion of a free diet. However, a recent study has shown that PEN can be effective in combination with a “*prudent pattern diet*” [61]. In this study, PEN, with an exclusion diet based on components hypothesized to affect the microbiome or intestinal permeability, was effective for induction of remission in children and young adults with CD. Several randomized controlled trials are being developed in order to confirm this observation (NCT01728870, NCT 02843100, NCT02231814).

The idea of maintaining remission with diet is compelling and challenging at the same time. On the one hand, once remission is established, it is certainly possible that diet will be sufficient to maintain homeostasis and to prevent the cascade of proinflammatory events leading to flare-ups; however, long-term studies are challenging to perform, and adherence becomes a serious concern. Long-term studies on the maintenance of remission have largely focused on the use of semi-elemental EEN [62,63]. These data show promise for the use of semi-elemental EEN in the maintenance of CD. On the other hand, an interesting idea has been revealed by a recent study showing that, in patients with moderate to severe CD undergoing infliximab treatment (anti-TNFα therapy), combined EEN therapy of ≥600 kcal/day led to an increase in the remission maintenance rate, preventing loss of response to infliximab therapy [64]. These studies illustrate the potential benefit of elemental nutrition in the maintenance of remission in patients with CD; however, there are as yet no data to support this hypothesis.

Regarding fatty acids, PUFAs comprise two main groups of fats: omega-3 and omega-6 fatty acids. Omega-6 is considered an inflammatory fat, whereas omega-3 is anti-inflammatory. In human IBD, a systematic review has not supported the notion that supplementation of omega-3 fatty acids can induce and maintain remission of IBD [65]. The latest Cochrane review has reached a similar conclusion [66]. Interestingly, several studies have demonstrated that different genotypes can be associated with the variable response to nutritional intervention with omega-3 PUFAs. For example, genetic polymorphisms of TNFα and PPARγ have been associated with an altered response to nutritional intervention with omega-3 PUFAs [67]. In IBD, genetic polymorphisms, such as CYP4F3 and caspase 9+93C/T, modify the association between dietary fatty acid intake and risk of IBD [68]. Thus, specific mutations should also be considered before considering dietary therapeutic interventions.

## 6. Conclusions

Apart from EEN in specific situations, no strong evidence has been found to provide solid dietary intervention recommendations for patients with IBD. However, evidence and knowledge are accumulating and have suggested that pertinent studies should be prospectively developed in order to characterize potential dietary therapeutic possibilities. This body of knowledge leads us to recommend that a “prudent” diet or a “Mediterranean diet” should be advised to patients. Given that diet has been shown to be important for intestinal homeostasis and to avoid disruption of intestinal permeability, the consumption of vegetables, fruits, olive oil, fish, grains and nuts should be recommended, together with avoidance of the use of excessive additives and industrialized food processing.

Dietary recommendations for patients with IBD should emerge in a personalized manner, taking into consideration nutrient knowledge, the individual mucosa–microbiome–immunity interplay, the status of epigenetic changes and creeping fat metabolism. Dietary recommendations cannot be generalized for all patients, however. Given that the above elements differ per patient, no single dietary intervention can offer the same improvement for all patients. Thus, future studies should take this factor into account and be designed accordingly.

## Figures and Tables

**Figure 1 nutrients-11-01062-f001:**
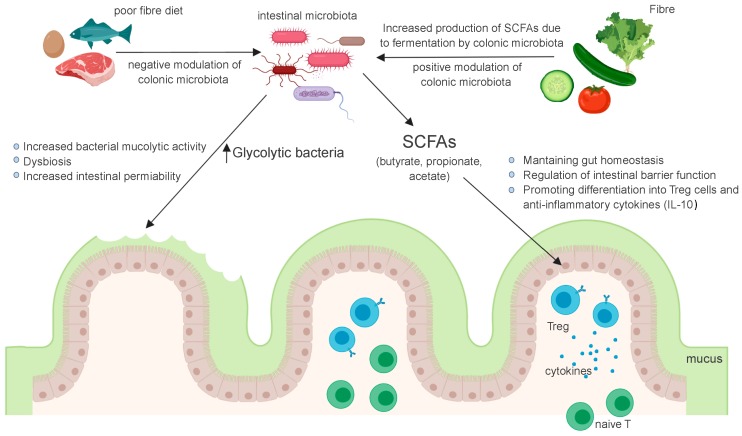
Interaction of dietary fiber with the gut microbiota.

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
