# Peer review of "Bases for the Adequate Development of Nutritional Recommendations for Patients with Inflammatory Bowel Disease"

_nutrients, 2019, doi:10.3390/nu11051062_

Round 1

Reviewer 1 Report

The authors have selected a very pertinent topic; research in which is somewhat lacking, and created a compelling argument for the need for further research. There is a need for English and grammar to be checked by a native English speaker; the title for example reads 'Bases for the adequate development....' but I suspect the writers mean 'Basis for adequate development...'.

Author Response

Response to Reviewer 1

The authors have selected a very pertinent topic; research in which is somewhat lacking, and created a compelling argument for the need for further research. There is a need for English and grammar to be checked by a native English speaker; the title for example reads 'Bases for the adequate development....' but I suspect the writers mean 'Basis for adequate development...'.

-    We are very grateful for all the suggestions that you have done to improve our manuscript.

-    We have made a great effort according to the suggestions. In addition, the manuscript has been reviewed and corrected by ServingMed.com, a professional, native, English-language medical editing service.

-       The title of the manuscript has been revised by a native English speaker and this was his recommendation: Bases for the Adequate Development of Nutritional Recommendations for Patients with Inflammatory Bowel Disease.

-      All changes have been highlighted in the text.

Reviewer 2 Report

I read with interest the MS "Bases for the adequate development of nutritional recommendations for patients with Inflammatory Bowel Disease (IBD)". by Esteban Sáez-González and coworkers. It is a well designed review paper dealing with a demanding issue. However, diet advise for IBD needs to be clearly distinguished between active and inactive disease. In addition, Crohn disease is commonly considered to be more susceptible than ulcerative colitis to diet advise and I would like to have the MS give some hints about it. Finally, I would suggest to add "for adult patients with IBD" to the title for elemental diet is effective treatment for pediatric Crohn disease

Author Response

Response to Reviewer 2:

I read with interest the MS "Bases for the adequate development of nutritional recommendations for patients with Inflammatory Bowel Disease (IBD)". by Esteban Sáez-González and coworkers. It is a well designed review paper dealing with a demanding issue. However, diet advise for IBD needs to be clearly distinguished between active and inactive disease. In addition, Crohn disease is commonly considered to be more susceptible than ulcerative colitis to diet advise and I would like to have the MS give some hints about it. Finally, I would suggest to add "for adult patients with IBD" to the title for elemental diet is effective treatment for pediatric Crohn disease

-    We are very grateful for all the suggestions that you have done to improve our manuscript. We have made a great effort according to the suggestions. In addition, the manuscript has been reviewed and corrected by ServingMed.com, a professional, native, English-language medical editing service.

-       According with the suggestions, in the “Diet as Possible Therapy” section of the manuscript, we have distinguished between active and inactive patients. Moreover, we have added new specific hints about Crohn’s disease diet. These changes have been highlighted in red in the text (Lines 722 to 727, Lines 839-841, Lines 847-848). The references 62-64 have been added.

-      “for adult patients with IBD” has been added to the title for elemental diet is effective treatment for pediatric Crohn diseases as suggested (line 724).

Reviewer 3 Report

The review article by Sáez-González and López-Muñoz et al provides comprehensive summary of the impact of nutrients in ulcerative colitis and crohn’s disease.  

Strengths.

The review article is well written and provides a summary of pre-clinical and clinical studies studying the impact of nutrients in IBD. They provide certain recommendations about dietary interventions like “prudent” diet or a “Mediterranean” diet intervention.  

Lastly authors mention about personalized treatment for each patient and talk about epigenetic considerations to be taken into account. However they cite only one article talking about epigenetic considerations and Mediterranean diet. I would recommend to add more pre-clinical and clinical studies determining the role of epigenetics in IBD. Moreover the feasibilty of all the recommendations taken into account should be considered or determined

Minor changes

1)      Effects of Nutrients in intestinal permeability and in microbiome and in immunity. Please change this title to something like Impact of nutrients on intestinal permeability, microbiome and immunity.

2)      The line 238 needs grammar proof which “PEN can be effective if the diet allows is pa “prudent pattern one”. As for this title “Other factors to take into account: The fat in the mesentery and epigenetic considerations”. I would recommend this title to change to “Mesenteric Fat and epigenetic considerations”

3)      Moreover, the sentence which starts with These days’ patients with IBD…. Should be corrected grammatically

4)      Line 215 should be changed grammatically.

5)      Some of the reference are cited in ( ) while others in [ ], I recommend to change to either to maintain uniformity.

6)      Figure should be labelled as figure 1. Also it says anti-inflammatory cytokines like IL-18

In fact IL-18 is a pro-inflammatory cytokine and is deleterious in colitis as given by following article’

Nowarski et al “Epithelial IL-18 Equilibrium Controls Barrier Function in Colitis” Cell (2015)

Author Response

Response to Reviewer 3

Strengths.

The review article is well written and provides a summary of pre-clinical and clinical studies studying the impact of nutrients in IBD. They provide certain recommendations about dietary interventions like “prudent” diet or a “Mediterranean” diet intervention.  

Lastly authors mention about personalized treatment for each patient and talk about epigenetic considerations to be taken into account. However they cite only one article talking about epigenetic considerations and Mediterranean diet. I would recommend to add more pre-clinical and clinical studies determining the role of epigenetics in IBD. Moreover the feasibilty of all the recommendations taken into account should be considered or determined

-We are very grateful for all the suggestions that you have done to improve our manuscript. We have added more pre-clinical and clinical studies determining the role of epigenetics in IBD (Lines 695-698; Lines 705-712). References 50-55 have been added.

Minor changes

1)      Effects of Nutrients in intestinal permeability and in microbiome and in immunity. Please change this title to something like Impact of nutrients on intestinal permeability, microbiome and immunity.

- The title section has been changed as suggested (Line 345)

2)      The line 238 needs grammar proof which “PEN can be effective if the diet allows is pa “prudent pattern one”. As for this title “Other factors to take into account: The fat in the mesentery and epigenetic considerations”. I would recommend this title to change to “Mesenteric Fat and epigenetic considerations”

- The sentence “PEN can be effective if the diet allows is pa prudenr pattern one” has been changed in the manuscript (Lines 842-843).

3)      Moreover, the sentence which starts with These days’ patients with IBD…. Should be corrected grammatically

- The manuscript has been reviewed and corrected by ServingMed.com, a professional, native, English-language medical editing service. The changes have been marked in the text.

4)      Line 215 should be changed grammatically.

- The manuscript has been reviewed and corrected by ServingMed.com, a professional, native, English-language medical editing service. The changes have been marked in the text.

5)      Some of the reference are cited in ( ) while others in [ ], I recommend to change to either to maintain uniformity.

- These references have been changed to maintain uniformity.

6)      Figure should be labelled as figure 1. Also it says anti-inflammatory cytokines like IL-18

In fact IL-18 is a pro-inflammatory cytokine and is deleterious in colitis as given by following article’

Nowarski et al “Epithelial IL-18 Equilibrium Controls Barrier Function in Colitis” Cell (2015)

-       Figure has been labelled as Figure 1.

-       IL-18 has been deleted from figure 1. Thanks for your comments.

Round 2

Reviewer 2 Report

I read with interest the revised version of the MS "Bases for the Adequate Development of Nutritional Recommendations for Patients with Inflammatory Bowel Disease" by Saez-Gonzalez E and coworkers. The MS has been improved and all of my queries have been addressed in full. No additional suggestions on this side.